# Clinical Features and Pathogenic Mechanisms of Gastrointestinal Injury in COVID-19

**DOI:** 10.3390/jcm9113630

**Published:** 2020-11-11

**Authors:** Keiichi Mitsuyama, Kozo Tsuruta, Hidetoshi Takedatsu, Shinichiro Yoshioka, Masaru Morita, Mikio Niwa, Satoshi Matsumoto

**Affiliations:** 1Inflammatory Bowel Disease Center, Kurume University Hospital, 67 Asahimachi, Kurume, Fukuoka 830-0011, Japan; tsuruta_kouzou@med.kurume-u.ac.jp (K.T.); takedatsu_hidetoshi@med.kurume-u.ac.jp (H.T.); yoshioka_shinichirou@kurume-u.ac.jp (S.Y.); morita_masaru@med.kurume-u.ac.jp (M.M.); 2Division of Gastroenterology, Department of Medicine, School of Medicine, Kurume University, 67 Asahimachi, Kurume, Fukuoka 830-0011, Japan; 3Institute for Advanced Sciences, 1-1-1 Umezono, Tsukuba, Ibaraki 305-8560, Japan; mikio_niwa@mail.toagosei.co.jp; 4Yakult Central Institute for Microbiological Research, Kunitachi, Tokyo 186-0011, Japan; satoshi-matsumoto@yakult.co.jp

**Keywords:** ACE2, COVID-19, gastrointestinal tract, intestine, gut microbiota, renin-angiotensin-aldosterone system, SARS-CoV-2, TMPRSS2, thrombosis, tryptophan

## Abstract

Severe acute respiratory syndrome coronavirus 2 (SARS-CoV-2) is the cause of the global coronavirus disease 2019 (COVID-19) outbreak. Along with the respiratory tract, the gastrointestinal (GI) tract is one of the main extra-pulmonary targets of SARS-CoV-2 with respect to symptom occurrence and is a potential route for virus transmission, most likely due to the presence of angiotensin-converting enzyme 2. Therefore, understanding the mechanisms of GI injury is crucial for a harmonized therapeutic strategy against COVID-19. This review summarizes the current evidence for the clinical features of and possible pathogenic mechanisms leading to GI injury in COVID-19.

## 1. Introduction

Severe acute respiratory syndrome coronavirus 2 (SARS-CoV-2) is the cause of the current coronavirus disease 2019 (COVID-19) pandemic. The pathogenic mechanisms underlying COVID-19, particularly with respect to multi-organ dysfunction, are not yet completely understood [1,2]. Emerging epidemiological data about COVID-19 suggest an association between gastrointestinal (GI) injury and SARS-CoV-2 infection in terms of clinical features, prognosis, and disease severity [3,4]. Although respiratory transmission and symptoms are the primary route and presentation of COVID-19, respectively, the GI system could be an alternative or additional route for virus transmission and clinical manifestation, most likely due to the distribution of angiotensin-converting enzyme 2 (ACE2) throughout the GI tract [5,6,7,8]. 

Therefore, the mechanisms underlying GI injury in COVID-19 patients need to be actively explored to develop strategies for mitigating the spread of SARS-CoV-2. However, so far, limited information is available concerning the mechanisms underlying COVID-19-associated GI injury. Here, we summarize current evidence for the clinical features and possible pathogenic mechanisms leading to GI injury in COVID-19. 

## 2. ACE2-Mediated Viral Entry

SARS-CoV-2 is transmitted via the respiratory system through saliva droplets or nasal discharge [2]. SARS-CoV-2 has also been suggested to be transmitted through the GI system as swallowed saliva or via the consumption of contaminated food, although this route of transmission may account for only a small proportion of COVID-19 cases [4,6]. The cellular entry of SARS-CoV-2 requires binding of the SARS-CoV-2 spike protein to the membrane-bound form of ACE2, a receptor on the cell surface that controls the cleavage of several peptides, on the target cell; this is followed by the priming of the spike protein by the host transmembrane serine protease 2 (TMPRSS2) to facilitate cell entry [9,10,11]. ACE2 and TMPRSS2 are expressed in several cell types other than lung alveolar cells, including the GI epithelium and vascular endothelium [5,6,12]. Importantly, the expression levels of these proteins, particularly ACE2, are much higher in the GI tract than in the lungs [7,13]. Attachment of the virus to ACE2 triggers the internalization of the ACE2-virus complex into the target cell, leading to the downregulation of ACE2 [14]. SARS-CoV-2 RNA is then released into the cytoplasm, and viral replication is initiated.

ACE2 is also present in the circulation in a soluble form, albeit at low levels [15]. Soluble ACE2, which lacks the cytosolic and transmembrane domains, is cleaved from full-length ACE2 on the cell membrane by a disintegrin and metalloproteinase (ADAM) 17 and released into the extracellular environment [16,17,18,19]. Soluble ACE2 is enzymatically active and blocks the binding of the spike protein to its receptor, indicating that it competitively inhibits viral entry into target cells. This mechanism provides the virus with “false” ACE2 and leaves the membrane-bound ACE2 free to perform its physiological tasks [20,21,22]. 

Notably, the expression of ACE2 or TMPRSS2 does not solely determine the course of COVID-19. The body’s immune response, the viral load, and some unidentified receptors may independently or jointly affect disease progression and prognosis.

## 3. Clinical GI Features

The clinical and experimental evidence discussed in the forthcoming sub-sections strongly suggests that SARS-CoV-2 in swallowed saliva or in contaminated food could reach the intestine to actively infect, as well as replicate in, the intestinal cells.

### 3.1. GI Symptoms

Clinical studies on patients with COVID-19 have shown that GI symptoms such as anorexia, nausea/vomiting, abdominal pain, and diarrhea appear to precede or follow pulmonary symptoms, with an incidence ranging from approximately 10 to 60% [23,24,25,26,27]. Studies reporting an association between GI symptoms and disease severity/susceptibility to COVID-19 are controversial. Some studies suggest that GI symptoms are associated with milder disease and a better prognosis [28]. In contrast, the results of a systemic review and meta-analysis showed that abdominal pain [23] or diarrhea [24,25,26,27] may be a risk factor for severe COVID-19. Interestingly, an animal study reported that intragastric inoculation of SARS-CoV-2 causes productive infection and leads to pathological changes in the lungs, suggesting that GI infection could contribute to pulmonary injury [29]. Therefore, early identification of GI symptoms is crucial, as, in some patients, this can precede the development of pulmonary symptoms. In contrast, some patients present with only GI symptoms and no pulmonary symptoms.

### 3.2. SARS-CoV-2 in the Feces

Emerging evidence has indicated that SARS-CoV-2 RNA can be detected via real-time reverse-transcription polymerase chain reaction (RT-PCR) in fecal samples for a long period, even after the pulmonary symptoms have resolved [24,30]. However, as RT-PCR may detect residual viral genome, it remains unclear whether this represents active viral replication in the tissues or a previous infection. More recently, independent laboratories reported the successful isolation of live SARS-CoV-2 from fecal samples of COVID-19 patients [31,32], suggesting that the presence of this virus in feces is the cause of GI symptoms and fecal–oral infection, rather than a mere bystander effect. This knowledge is important for containing viral spread. It should be noted that GI involvement leads to higher viral load and/or prolonged viral shedding [33], allowing the virus to disseminate to other organs, including the lungs, liver, and kidneys. GI tropism of SARS-CoV-2 is further supported by the evidence of elevated fecal levels of calprotectin, a protein marker for intestinal inflammation [34].

### 3.3. SARS-CoV-2 in GI Tissues

Studies based on single-cell RNA sequencing have confirmed the expression of ACE2 and TMPRSS2 in the GI epithelium [35]. Xiao et al. performed GI biopsy during endoscopy on patients positive for SARS-CoV-2 in the fecal samples and demonstrated that the cytoplasm of the epithelium lining the GI tract, including the stomach, duodenum, and rectum, tested positive for ACE2 and viral nucleocapsid protein. These results suggest that SARS-CoV-2 invades the mucosal cells of the stomach and the intestine and produces infectious virions [36]. Importantly, active replication of SARS-CoV-2 was also demonstrated by the experimental infection of human small intestinal organoids [37,38].

### 3.4. SARS-CoV-2 and the Gastric Acid Barrier

Although gastric acid can significantly reduce virus viability, SARS-CoV-2 is not inactivated in the stomach at gastric pH higher than 3.0 at room temperature [39,40]. This suggests the possibility that the virus travels to the intestine by passing through the gastric acid barrier, at least under conditions of chronic gastritis or *Helicobacter pylori* infection, the use of gastric acid inhibitors, or previous gastrectomy [41]. However, to date, several studies exploring the association between the use of proton pump inhibitors and increased disease severity/susceptibility to COVID-19 are controversial [42,43,44], making it difficult to determine whether there is indeed an increased risk for SARS-CoV-2 infection and COVID-19-related death for users of proton pump inhibitors. Future studies are needed to support or counter this association.

### 3.5. GI Imaging

Some studies used abdominal diagnostic imaging in patients with SARS-CoV-2 infection to assess their intestinal condition and provide evidence for GI involvement. The most frequently reported observations are intestinal ischemia, including vessel thrombosis/embolism in the small bowel and/or colon and ischemic colitis [45,46,47,48,49,50,51]. Bhayana et al. examined 42 abdominal CT scans of 412 hospitalized patients with COVID-19 [48]. They identified bowel wall abnormalities, including pneumatosis and portal venous gas suggestive of ischemia, in 31% of the CT scans, mostly in intensive care unit patients. Laparotomy and pathology findings confirmed small bowel ischemia in some patients, which may have been due to small vessel thrombosis, confirming that bowel abnormalities, including ischemia, are common findings among critically ill COVID-19 patients; however, the cause of intestinal abnormalities in patients who did not undergo surgery remains unclear.

The GI injury observed in COVID-19 is not limited to intestinal ischemia because cases without evidence of ischemia have also been reported [46,52,53,54,55]. In a case report of a SARS-CoV-2 GI infection causing acute hemorrhagic colitis, as diagnosed by colonoscopy, the biopsy showed slight expansion of the lamina propria by edema, with normal cellularity, intact crypts, and no evidence of ischemia, virocytes, or protozoa [52]. Massironi et al. identified endoscopic lesions in 70% (14/20) of colonoscopies on patients with COVID-19 [46]. The main findings, other than four cases (20%) of histologically confirmed colon ischemia, included five cases (25%) of segmental colitis associated with diverticulosis and one case of diffuse hemorrhagic colitis; in three cases, the colonic mucosa appeared normal on visualization, but histological evidence of microscopic (two cases) and lymphocytic (one case) colitis was found. However, it is unclear whether the patients developed those lesions after SARS-CoV-2 infection or if these symptoms were already present in these patients before they contracted COVID-19 and were then exacerbated by the disease. At present, the specificity of these imaging findings to COVID-19 is unclear. Further investigations are needed to clarify the characteristics of the diagnostic images of GI injury in COVID-19.

## 4. Pathogenic Mechanisms of GI Injury

GI injury in patients with COVID-19 is likely to have several etiological factors. Pathologically, this injury can be divided into two groups: primary GI injury, in which SARS-CoV-2 transmits to the GI tract by passing through the digestive system, and secondary GI injury, associated with pulmonary SARS-CoV-2 infection, in which the virus transmits by passing through the respiratory system (Figure 1). Primary GI injury is caused by direct cytotoxic damage, dysregulation of the renin–angiotensin–aldosterone system (RAAS), or malabsorption of tryptophan in the intestinal epithelium. Secondary GI injury is caused by endothelial damage and thrombo-inflammation in the blood vessels or dysregulation of the immune system with systemic circulation. Gut dysbiosis could also be responsible for primary or secondary GI injury. These factors can affect each other and exacerbate GI injury. Such GI injury would likely lead to a higher viral load and/or prolonged viral shedding, allowing the virus to disseminate to other organs.

### 4.1. Direct Virus-Mediated Cytotoxic Damage in the Intestinal Epithelium

The abundance of ACE2 in the GI epithelium makes the GI tract vulnerable to SARS-CoV-2 entry. Once the virus enters the epithelium, direct cellular damage, possibly through pyroptosis and apoptosis, might also actively contribute to injury and inflammation of the GI epithelium [8,56]. Moreover, this is supported by the intracellular staining of viral nucleocapsid protein in almost the entire GI tract [57]. At present, it is unclear whether SARS-CoV-2 infection can directly cause GI injury or if it simply triggers an overwhelming host response, which could secondarily result in COVID-19-associated GI dysfunction.

### 4.2. Dysregulation of the RAAS in the Intestinal Epithelium

ACE2 is a negative regulatory enzyme in the RAAS. In addition to its function as a receptor for viral entry, it influences immune functions. The components of the RAAS are characteristic of the GI system, as well as of the cardiovascular or renovascular system [58]. In the RAAS, ACE cleaves angiotensin I (Ang I) to Ang II, which binds to the Ang type 1 receptor (AT1R) and mediates numerous systemic and local processes, including vasoconstriction, inflammation, fibrosis, and thrombosis. In contrast, ACE2 catalyzes the conversion of Ang II to Ang (1–7). Ang (1–7) binds to the G protein-coupled receptor Mas to mediate vasodilatation, as well as anti-inflammatory, anti-fibrotic, and anti-thrombotic effects. As Ang II/AT1R signaling also promotes an immune response, ACE2 might control immune functions via the Ang (1–7)/Mas axis. The balance between the “adverse” ACE/Ang II/AT1R axis and the “protective” ACE2/Ang (1–7)/Mas axis determines the overall homeostatic effect of the RAAS [59].

Functional studies based on colitis animal models have indicated that the modulation of ACE2 expression affects the severity of intestinal inflammation. ACE2 deficiency causes enhanced susceptibility to dextran sodium sulfate-induced colitis [60], suggesting that ACE2 plays a protective role in colitis. Moreover, Ang (1–7) treatment alleviates colitis progression, whereas the blockade of Mas aggravates the disease [61], indicating the protective role of the ACE2/Ang (1–7)/Mas axis. In contrast, treatment with the ACE2 inhibitor GL1001 reduces the severity of colitis [62], suggesting that ACE2 plays a pathogenic role in intestinal inflammation. During SARS-CoV-2 infection, the downregulation of ACE2 would potentially result in unopposed functions of Ang II and decreased levels of Ang (1–7), thereby shifting the balance towards the pro-inflammatory side, which would contribute to epithelial injury [63,64].

### 4.3. Malabsorption of Tryptophan in the Intestinal Epithelium

ACE2 also exhibits an RAAS-independent effect as a regulator of amino acid transport in the intestine [65]. ACE2 binds the membrane-bound amino acid transporter B0AT1 as a chaperone and contributes to the absorption of tryptophan, an essential amino acid with a critical role in intestinal homeostasis [66,67]. Dietary tryptophan is absorbed via the B0AT1/ACE2 interaction on the luminal surface of the small intestinal epithelium. This results in the activation of the mammalian target of rapamycin (mTOR) directly through nutrient sensing and/or through the tryptophan–nicotinamide pathway. mTOR in turn regulates the expression of antimicrobial peptides that affect the composition of the gut microbiota [60].

A lack of ACE2 accentuates GI susceptibility to inflammation. ACE2-knockout mice do not express B0AT1 in their small intestine and therefore exhibit decreased tryptophan uptake and the subsequent activation of mTOR. This leads to decreased expression levels of antimicrobial peptides and alteration in the gut microbiota, resulting in increased sensitivity to intestinal inflammation [60]. Based on the available literature, it is suggested that SARS-CoV-2 binds to ACE2 in the intestinal epithelium, resulting in a disturbance in ACE2 functions, which leads to GI injury.

### 4.4. Endothelial Damage and Thrombo-Inflammation in Blood Vessels

ACE2 is highly expressed in the arterial and venous endothelium, suggesting that blood vessels may also be susceptible to SARS-CoV-2 infection [12]. The importance of the endothelium in COVID-19 pathogenesis was emphasized by recent histopathological data that reported direct endothelial infection, endothelial inflammation, and microvascular and macrovascular thrombosis in venous and arterial circulation, along with inflammatory cell accumulation and endothelial and inflammatory cell death [68]. This suggests that SARS-CoV-2 infection facilitates the induction of endothelial inflammation as a direct consequence of viral involvement, host inflammatory response, and possibly apoptosis and pyroptosis [69].

Endothelial activation/damage due to virus binding to ACE2, and the promotion of inflammation and hyper-coagulation may explain the high thrombotic burden observed in COVID-19. This depends on the complex interplay between increased endothelial dysfunction/damage, pro-inflammatory cytokine release, and potential coagulopathy in severe disease, which collectively promote the activation of coagulation. This can trigger excessive thrombin production, inhibit fibrinolysis, and activate complement pathways, thereby initiating thrombo-inflammation that ultimately leads to microthrombi deposition and microvascular dysfunction in different organs, including the GI system [70].

Neutrophil extracellular traps (NETs) are networks of extracellular fibers composed of DNA that contain histones and granular proteins, such as myeloperoxidase and elastase, which may provide a template for trapping platelets and for thrombus formation in the microcirculation [71]. NETs induce tissue injury by exerting a direct toxic effect on endothelial cells; this toxicity may be a result of the high local concentration of histones and granular proteins [72]. Recent evidence indicates that NET formation is enhanced in the neutrophils of patients with COVID-19 [73,74]. Together with the in vitro data, which demonstrated that viable SARS-CoV-2 can directly induce the release of NETs by healthy neutrophils [75], this shows that an exaggerated level of NET release induced by the virus may contribute to COVID-19-associated tissue damage, thrombosis, and fibrosis.

### 4.5. Dysregulation of the Immune System

SARS-CoV-2 disrupts normal immune responses, leading to an impaired immune system and uncontrolled inflammatory responses in patients with COVID-19. SARS-CoV-2 not only activates antiviral immune responses but also causes uncontrolled inflammatory responses that are characterized by the activation of systemic and local cytokine-secreting cells with innate and adaptive immune mechanisms [76,77].

The immunologic reactions in severe COVID-19 lead to the cytokine storm, which is an out-of-control cytokine release, thereby creating a hyperinflammatory condition in the host [78]. A cytokine storm is induced by the activation of a large number of leukocytes, including B cells, T cells, monocytes, neutrophils, and other resident cells, such as those in the epithelium and endothelium, which release large numbers of pro-inflammatory cytokines [5,79,80,81,82]. In such a setting, the integrity of the epithelial and endothelial barrier in nearby tissues may be weakened, thereby damaging multiple organ systems, including those of the GI system. In the intestine, cytokine storm may contribute to the activation of the mucosal immune system, which not only enhances an immune-mediated inflammatory process but also affects the gut microbiota, leading to GI injury [8,83].

SARS-CoV-2 also leads to lymphopenia, lymphocyte activation and dysfunction, and granulocyte and monocyte abnormalities, which cause a defect in antiviral response and immune regulatory function of these cells [84]. Lymphopenia, a marker of impaired cellular immunity, is a cardinal laboratory finding in patients with COVID-19, with a prognostic association in the vast majority of published studies [85]. Patients also show marked reductions in the counts of CD4^+^ and CD8^+^ T cells, natural killer cells, and B cells [77]. During severe COVID-19, the GI system faces widespread inflammation and tissue damage caused by these events [86].

### 4.6. Gut Dysbiosis

It is well established that the gut microbiota plays a critical role in intestinal homeostasis and that changes in its composition are involved in intestinal inflammation [87]. Several lines of evidence suggest that SARS-CoV-2 infection is associated with alterations in the gut microbiota [88,89,90]. Zuo et al. provided evidence of prolonged gut dysbiosis, characterized by the enrichment of pathogenic microbes and opportunistic pathogens, along with the depletion of beneficial commensals in critically ill patients with COVID-19, and its association with disease severity [88]. The gut dysbiosis observed in these studies is multifactorial and involves epithelial dysfunction (especially impaired antimicrobial peptide production that controls the gut microbial community) [66,67], as well as immune cell dysfunction (especially a cytokine storm) [78]. To date, it is not clear whether such dysbiosis is specific to SARS-CoV-2 infection or a consequence of critical illness. Further investigations based on larger cohorts are needed to determine whether alterations in the gut microbiota influence the GI and pulmonary outcomes of COVID-19.

If large studies confirm that the gut microbiota indeed affects the severity of COVID-19, targeting the microbiota may be an attractive therapeutic strategy. At present, no direct clinical evidence suggests that modulation of the gut microbiota has therapeutic value in patients with COVID-19. However, it is tempting to speculate that targeting the gut microbiota through probiotics, prebiotics, and fecal microbiota transplantation might be a potential therapeutic strategy [8].

### 4.7. Others

Hypoxia is a major symptom in patients with COVID-19 [91] and is also critical for intestinal homeostasis, including microbiota composition and function [92]; thus, oxygen deprivation may be important in the context of GI injury. The enteric nervous system could be affected by SARS-CoV-2, either via direct viral infection or by the elicited immune response. It is possible that alterations in the enteric nervous system may further aggravate GI injury [8].

## 5. SARS-CoV-2 and Liver Injury

ACE2 is expressed in cholangiocytes and hepatocytes (20 times higher in cholangiocytes than hepatocytes), although to a lesser degree than lung alveolar cells and the GI epithelium [93]. Therefore, up to 50% of COVID-19 patients experience various degrees of liver function abnormalities [94,95]. Since SARS-CoV-2 enters the mucous membranes in the intestine, it may access the biliary system through the portal vein and bind to ACE2-positive cholangiocytes and hepatocytes, resulting in direct damage to the liver (cytopathic effect) [96,97]. The hyper-inflammation seen with the cytokine storm, hypoxia-associated metabolic derangements, and gut dysbiosis are other potential mechanisms of liver damage [98,99]. Drug-induced liver injury, particularly secondary to antivirals and antibiotics, is also an important consideration [100].

At present, the impact of SARS-CoV-2 on chronic liver diseases remains largely unknown. A previous study has shown that patients with pre-existing hepatitis B infection had more severe courses of COVID-19 [101]. Patients with liver cirrhosis have also been reported to be more susceptible to SARS-CoV-2 infection, probably due to their immunocompromised status [102,103]. The impact of SARS-CoV-2 on inappropriate alcohol consumption is under investigation [104]. The potential adverse effects of viral infection on the prognosis of patients with chronic liver diseases need further investigation.

## 6. Conclusions and Future Directions

We are slowly beginning to understand the complex pathogenesis of SARS-CoV-2 infections. The widespread organ-specific complications of COVID-19, including those of the GI system, are now increasingly being appreciated. An in-depth understanding of the GI injury and clinical manifestations of this multi-organ disease remains imperative. The GI system could serve as a SARS-CoV-2 entry site and a potentiating organ due to the widespread presence of ACE2 and TMPRSS2 in this system, which may be involved in magnifying the systemic inflammatory response. Further studies are warranted to provide novel insights into the GI system as a potential virus entry point or a potentiator of infection, which will be pivotal for developing a harmonized therapeutic strategy against GI injury in COVID-19.

## Figures and Tables

**Figure 1 jcm-09-03630-f001:**
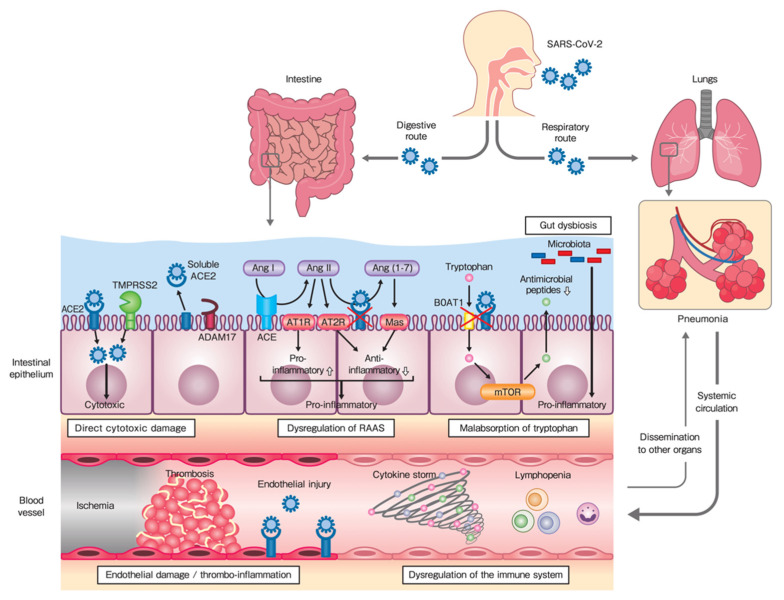
Proposed mechanisms of gastrointestinal (GI) injury in coronavirus disease 2019 (COVID-19): primary GI injury, in which SARS-CoV-2 transmits through the digestive route, and secondary GI injury—associated with pulmonary infection—in which the virus transmits through the respiratory route. Gut dysbiosis could also be responsible for primary or secondary GI injury. SARS-CoV-2, Severe acute respiratory syndrome coronavirus 2; ACE, angiotensin-converting enzyme; TMPRSS, transmembrane serine protease; ADAM, a disintegrin and metalloproteinase; Ang, angiotensin; AT1R, angiotensin type 1 receptor; AT2R, angiotensin type 2 receptor; RAAS, renin-angiotensin-aldosterone system; mTOR, mammalian target of rapamycin.

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
