# Peer review of "Clinical Features and Pathogenic Mechanisms of Gastrointestinal Injury in COVID-19"

_jcm, 2020, doi:10.3390/jcm9113630_

Round 1
Reviewer 1 Report
This is very interesting review that concerns a very relevant topic.
The part of the text related to the pathogenetic mechanisms of the effects of covid on the GI system is explained in a clear and comprehensive way.
I think it is appropriate to discuss in a more exhaustive the section about the clinical gastroenterological aspects.
In particular, it is necessary to mention some recent studies that show how the presence of gastrointestinal symptoms, such as diarrhea and vomit, are associated with a better prognosis (see Aghemo et al).
Moreover, I think it is appropriate that the authors dedicate a part of the text to the effects of SARS-Cov2 on the liver. In particular, referring to potential adverse effects of viral infection on the prognosis of patients with advanced chronic liver disease.
Author Response
Response to Reviewer 1
This is very interesting review that concerns a very relevant topic. The part of the text related to the pathogenetic mechanisms of the effects of covid on the GI system is explained in a clear and comprehensive way.
#1. I think it is appropriate to discuss in a more exhaustive the section about the clinical gastroenterological aspects. In particular, it is necessary to mention some recent studies that show how the presence of gastrointestinal symptoms, such as diarrhea and vomit, are associated with a better prognosis (see Aghemo et al).
Response:According to the reviewer’s advice, we cited the report of Aghemo et al. and added the following description(page 2, lines 70-72): “Studies reporting an association between GI symptoms and disease severity/susceptibility to COVID-19 are controversial. Some studies suggest that GI symptoms are associated with milder disease and a better prognosis.”
#2. Moreover, I think it is appropriate that the authors dedicate a part of the text to the effects of SARS-Cov2 on the liver. In particular, referring to potential adverse effects of viral infection on the prognosis of patients with advanced chronic liver disease.
Response:According to the reviewer’s advice, we added a new section titled “SARS-CoV-2 and liver injury” on page 7, lines 279-294 We also referred to the potential adverse effects of SARS-CoV-2 infection on the prognosis of patients with chronic liver disease in this section.
Reviewer 2 Report
The authors review GI injury in COVID-19. The number of COVID-19 review articles is quickly starting to outnumber the number of works of original research. That said, this article is professionally written, well organized with clear sub-headings, and seems to fairly present what is not vs not known. Most of my comments are relatively minor and designed to avoid overstatements.
P 1 line 40-41. The statement that SARS-CoV-2 is transmitted through the GI tract needs to be modified. Most evidence suggests that while GI tract transmission is possible, viral viability on surfaces is low and therefore that GI transmission accounts for a small fraction of disease. The reference cited for this statement (Trottein) is another review article. In general, I encourage the authors to cite original studies instead of citing other reviews.
P 2 line 65-70. Many reports now suggest that GI symptoms are associated with milder disease (i.e., with better clinical outcomes) but with longer infections. These studies should be discussed.
P 3 line 96. SARS-CoV-2 is not inactivated at pH > 3. Is it inactivated at a normal physiologic gastric pH of 1-2?
P 3 line 99-101. The reports of an association between PPI use and increased disease severity/susceptibility to COVID are controversial and this statement should be toned down.
Section 2.5 GI Imaging. How specific are these imaging findings to COVID-19 versus critical illness generally? I suspect that most of these findings are non-specific for COVID-19.
P 4 line 150. By what mechanism does viral replication damage the GI epithelium? Does viral replication induce apoptosis? Generate ROS? Etc.
Section 3.4. This section doesn’t link endothelial damage to GI injury well, so it feels off topic.
Section 3.5. This seems like a missed opportunity. What specifically links the cytokine storm syndrome to GI injury/GI problems?
Section 3.6. Again, it’s not clear whether “dysbiosis” is specific to infection with SARS-CoV-2 or whether it is the consequence of critical illness. The latter seems more likely. Can the authors address this?
Author Response
Response to Reviewer 2
The authors review GI injury in COVID-19. The number of COVID-19 review articles is quickly starting to outnumber the number of works of original research. That said, this article is professionally written, well organized with clear sub-headings, and seems to fairly present what is not vs not known. Most of my comments are relatively minor and designed to avoid overstatements.
#1. P 1 line 40-41. The statement that SARS-CoV-2 is transmitted through the GI tract needs to be modified. Most evidence suggests that while GI tract transmission is possible, viral viability on surfaces is low and therefore that GI transmission accounts for a small fraction of disease. The reference cited for this statement (Trottein) is another review article. In general, I encourage the authors to cite original studies instead of citing other reviews.
Response: According to the reviewer’s suggestion, we changed the sentence from “SARS-CoV-2 is transmitted via the respiratory system through saliva droplets or nasal discharge, as well as through the GI system as swallowed saliva or via the consumption of contaminated food.” to “SARS-CoV-2 is transmitted via the respiratory system through saliva droplets or nasal discharge. SARS-CoV-2 has also been suggested to be transmitted through the GI system as swallowed saliva or via the consumption of contaminated food, although this route of transmission may account for only a small proportion of COVID-19 cases.” (Page 1, lines 40-43)
#2. P 2 line 65-70. Many reports now suggest that GI symptoms are associated with milder disease (i.e., with better clinical outcomes) but with longer infections. These studies should be discussed.
Response:According to the reviewer’s advice, we cited the report of Aghemo et al. and added the following descriptionon page 2, lines 70-72: “Studies reporting an association between GI symptoms and disease severity/susceptibility to COVID-19 are controversial. Some studies suggest that GI symptoms are associated with milder disease and a better prognosis.”
#3. P 3 line 96. SARS-CoV-2 is not inactivated at pH > 3. Is it inactivated at a normal physiologic gastric pH of 1-2?
Response: Thank you for this comment. Chin et al. examined the stability of SARS‐CoV‐2 at different pH conditions (ranging from pH 3 to pH 10) and found that SARS-CoV-2 was extremely stable under a wide range of pH values at room temperature. However, they did not examineits stability at the physiological gastric pH of 1-2. Accordingly, we did not mention this in our review. For your reference, SARS-CoV, but not SARS-CoV-2, is completely inactivated by highly acidic conditions (pH 1–3) at 37 °C, but moderate variations in pH conditions from 5 to 9 had minimal effect on the virus titer, regardless of the temperature (from 4 to 37 °C) [Darnell MER, et al. J Virol Methods 2004;121: 85–91]. Thus, we made a slight modification to the statement as follows (page 3, lines 103-104): “Although gastric acid can significantly reduce virus viability, SARS-CoV-2 is not inactivated in the stomach at gastric pH higher than 3.0 at room temperature.”
#4. P 3 line 99-101. The reports of an association between PPI use and increased disease severity/susceptibility to COVID are controversial and this statement should be toned down.
Response: As mentioned by the reviewer, studies reporting an association between PPI use and increased disease severity/susceptibility to COVID are controversial. Therefore, we changed the description (page 3, lines 107-111) to “However, to date, several studies exploring the associationbetween the use of proton pump inhibitorsand increased disease severity/susceptibility to COVID-19 are controversial,making it difficult to determine whether there is indeed an increased risk for SARS-CoV-2 infection and COVID-19-related death for users of proton pump inhibitors.Future studies are needed to support or counter this association.”
#5. Section 2.5 GI Imaging. How specific are these imaging findings to COVID-19 versus critical illness generally? I suspect that most of these findings are non-specific for COVID-19.
Response:Thank you for your valuable comment. We also agree that most of these findings might be non-specific for COVID-19; however, further collection of GI imaging in COVID-19 is needed. Therefore, we inserted the following sentence on page 3, lines 135-136: “At present, the specificity of these imaging findings to COVID-19 is unclear.”
#6. P 4 line 150. By what mechanism does viral replication damage the GI epithelium? Does viral replication induce apoptosis? Generate ROS? Etc.
Response: According to the reviewer’s advice, we added information regarding the possible mechanism underlying cellular damage as follows (page 4, lines 161-162): “Once the virus enters the epithelium, direct cellular damage, possibly through pyroptosis and apoptosis,might also actively contribute to injury and inflammation of the GI epithelium.” We also added an additional reference[Li S, et al.Clinical and pathological investigation of patients with severe COVID-19. JCI Insight. 2020. PMID: 32427582].
#7. Section 3.4. This section doesn’t link endothelial damage to GI injury well, so it feels off topic.
Response: We apologize that this was not clearly explained in the text. We changed the sentence from “This can trigger excessive thrombin production, inhibit fibrinolysis, and activate complement pathways, thereby initiating thrombo-inflammation that ultimately leads to microthrombi deposition and microvascular dysfunction.” to “This can trigger excessive thrombin production, inhibit fibrinolysis, and activate complement pathways, thereby initiating thrombo-inflammation that ultimately leads to microthrombi deposition and microvascular dysfunctionin different organs, including the GI system.” (Page 6, line 220)
#8. Section 3.5. This seems like a missed opportunity. What specifically links the cytokine storm syndrome to GI injury/GI problems?
Response: We apologize that this was not clearly explained in the text. We specifically described the onset of GI injury/GI problemsdue to the cytokine storm syndrome as follows (page 6, lines 242-245):“In the intestine, cytokine storm may contribute to the activation of the mucosal immune system, which not only enhances an immune-mediated inflammatory process but also affects the gut microbiota, leading to GI injury [8,83].” We also added the corresponding references.
#9. Section 3.6. Again, it’s not clear whether “dysbiosis” is specific to infection with SARS-CoV-2 or whether it is the consequence of critical illness. The latter seems more likely. Can the authors address this?
Response: As the reviewer mentioned, it is not clear whether “dysbiosis” is specific to SARS-CoV-2 infection or a consequence of critical illness. We added the following sentence on pages 6-7, lines 262-263 to clarify this: “To date, it is not clear whether such dysbiosis is specific to SARS-CoV-2 infection or a consequence of critical illness.”
Round 2
Reviewer 2 Report
No additional comments.
Author Response
Response to Reviewer
The paper is interesting and well-written.
#1. I suggest to improve the part included in the lines 281-295, adding to the reference 98
THE FOLLOWING: Gut Microbiota and Liver Interaction through Immune System Cross-Talk: A Comprehensive Review at the Time of the SARS-CoV-2 Pandemic.
Scarpellini et al. J Clin Med. 2020 Aug 3;9(8):2488.
Response:According to the reviewer’s suggestion, we added “gut dysbiosis”as one potential mechanism of liver damageand cited the new reference #99 (page 7, line 287).
#2. Furthermore, to highlight the potential relationship betrween COVID and etiology of liver disease, at line 289 it should be added that also inappropriate alcohol consumption is a field under investigation (ref: Alcohol consumption in the COVID-19 era. Testino G et al. Minerva Gastroenterol Dietol. 2020 Jun;66(2):90-92.)
Response:According to the reviewer’s suggestion, we added the sentence on the alcohol consumption and cited the new reference #104 (page 7, line 293).
In addition, we added the number 2, which we forgot to add in the previous version, prior to the heading entitled "ACE2-mediated viral entry" (page 1, line 39).